# LANGUAGE-GUIDED 4D GAUSSIAN SPLATTING FOR REAL-TIME DYNAMIC SCENE RENDERING

## ABSTRACT

Dynamic rendering methods often prioritize photometric fidelity while lacking explicit semantic representations, which constrains their ability to perform semantically guided rendering. To this end, we introduce Language-Guided 4D Gaussian Splatting (L4DGS), a lightweight framework for real-time dynamic scene rendering that integrates natural language into semantically structured 4D Gaussian representations. Central to L4DGS is a Sparse Multi-Scale Attention (SMSA) mechanism that enables fine-grained, language-driven control by emphasizing semantically relevant regions across space and time. To enforce semantic fidelity and spatial coherence, we propose a static regularization that aligns language-guided features with rendered outputs and ensures consistent depth. To further ensure temporal consistency, A dynamic regularization penalizes abnormal variations in semantics and depth over consecutive unit time intervals. L4DGS achieves a 16.1% improvement in PSNR, reduces perceptual error by 58.8%, and increases rendering speed by over 50%. Experimental results demonstrate the superiority of our approach in both visual quality and computational efficiency.

## 1 INTRODUCTION

Recent advances in neural rendering have enabled high-fidelity scene synthesis with remarkable visual realism Mildenhall et al. (2020); Pumarola et al. (2021); Park et al. (2021b); Li et al. (2022b); Gao et al. (2021). However, most existing approaches are designed for static settings and lack explicit semantic representations. This absence of semantic structure constrains their applicability in interactive, semantically guided, and dynamic environments Knapitsch et al. (2017); Hedman et al. (2018); Barron et al. (2022). Bridging the gap between human intent and real-time visual content creation is increasingly critical in computer graphics, impacting applications from semantic scene editing to immersive VR/AR, interactive media, and human-robot interaction. Addressing this challenge requires joint reasoning over space, time, and semantics, posing intensive demands on representation learning, cross-modal alignment, and temporally coherent rendering.

Recent advances in 3D Gaussian Splatting (3DGS) Kerbl et al. (2023) have demonstrated the effectiveness of point-based volumetric representations for real-time, photorealistic scene rendering. These approaches offer efficient rendering pipelines and high visual fidelity, making them well-suited for interactive graphics applications. In parallel, efforts to incorporate semantic understanding into 3D scene representations, through language, vision models, or segmentation-guided supervision, have enabled controllable generation and semantic editing. However, existing methods for dynamic scene rendering remain largely semantically agnostic, limiting their ability to align visual outputs with user intent or language-based descriptions. Moreover, current techniques fall short in addressing the challenge of maintaining semantic consistency over time, which is critical for rendering dynamic environments that evolve coherently. These limitations underscore the need for a unified framework that integrates language guidance into temporally consistent 4D representations, enabling semantically grounded, real-time rendering in dynamic and interactive scenarios.

Addressing language-guided rendering in dynamic scenes presents a set of fundamental research challenges. First, it requires learning a joint representation that aligns visual and linguistic modalities across both spatial and temporal dimensions, despite their inherently different structures and granularities. Achieving effective cross-modal alignment is challenging due to the semantic ambiguity of natural language and the limited spatial precision of pretrained vision-language models.

Second, incorporating language guidance into the rendering pipeline demands attention mechanisms that are both expressive and computationally efficient. These mechanisms should modulate scene content selectively and responsively, enabling real-time control without incurring excessive overhead. Third, the absence of explicit supervision for dynamic semantics complicates training and generalization, making it difficult to learn robust semantic representations over time. Furthermore, ensuring temporal coherence in dynamic scenes requires consistent modeling of object semantics across time. Separate-time-step supervision often leads to semantic drift, identity instability, or temporal artifacts such as flickering, especially under motion blur or sparse observations.

To address these challenges, we propose Language-Guided 4D Gaussian Splatting (L4DGS), a lightweight framework that integrates natural language understanding into real-time dynamic scene synthesis via semantically aware 4D Gaussian representations. Our design is motivated by a key observation: existing rendering pipelines lack the capacity to incorporate language guidance in a manner that is both spatially and temporally consistent. These methods largely render static scenes and are unable to capture the continuous evolution of semantics over time. L4DGS is built upon a sparse, multi-scale cross-modal attention mechanism that dynamically fuses language and visual features, guiding both the spatial placement and temporal progression of 4D Gaussian primitives. This core mechanism is complemented by a hierarchical regularization strategy, wherein both static and dynamic constraints are modulated by language-conditioned attention maps. These components enforce semantic consistency, geometric fidelity, and temporal coherence, enabling L4DGS to generate renderings that are not only photorealistic but also semantically aligned with user intent. This unified framework integrates high-level language-guided control with low-level dynamic scene rendering, ensuring real-time performance with scalability and computational efficiency.

To enable semantically guided rendering, we integrate natural language understanding with semantically aware 4D Gaussian representations. Leveraging a hierarchical semantic representation, our language-guided attention mechanism constrains the Gaussian primitives using object-aware features, ensuring that the rendered scene accurately reflects the semantics specified by the language input. In contrast to existing methods that depend on segmentation masks or external generators, our approach directly extracts fine-grained visual semantics from the input image using a center-differenced convolutional network. This network is enhanced with dilated convolutions to expand the receptive field without additional computational overhead, allowing for efficient context aggregation in high-resolution scenes. Building on these hierarchical semantics, we introduce a Sparse Multi-Scale Attention (SMSA) mechanism that adaptively aligns language with semantically and structurally salient visual regions. Rather than relying on dense attention across all tokens, SMSA incorporates a top-$k$ sparse attention strategy to focus the model's capacity on the most relevant spatial features, substantially improving both efficiency and interpretability.

To ensure spatial consistency, we further introduce a static regularization scheme that aligns language-guided visual features with rendered scene features in both magnitude and direction. This facilitates accurate correspondence between semantic features, such as object descriptions or action references, and the visual output. Furthermore, we introduce a static depth regularization term, modulated by language-conditioned attention, to preserve occlusion relationships, relative object positions, and the overall 3D scene geometry. To address temporal coherence, we further incorporate a dynamic regularization strategy tailored for dynamic scene rendering. Unlike static settings, dynamic scenes demand feature continuity over time to prevent artifacts such as flickering, motion blur, and semantic drift. Our approach extends both semantic and depth consistency across unit time intervals, rather than separate time steps, enabling robust alignment of features under fast motion, occlusions, and sparsely observed regions. This design ensures temporally stable and semantically meaningful rendering in complex, real-time dynamic environments. In summary, our contributions are as follows:

- The introduction of a novel framework that integrates language-guided semantics into dynamic scene rendering, addressing the gap between visual features and high-level semantics. To our knowledge, L4DGS is the first language-embedded real-time 4D rendering algorithm.

- A sparse multi-scale attention mechanism, leveraging language-guided attention to dynamically align language and visual features across multiple granularities, prioritizing semantically salient regions of the scene.

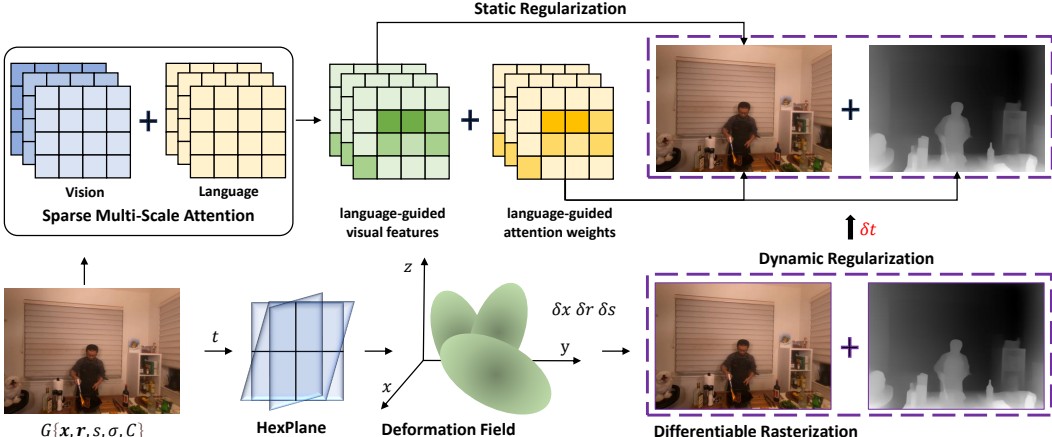

Figure 1: **Framework Overview.** F4DGS leverages a Sparse Multi-Scale Attention (SMSA) mechanism that integrates hierarchical visual features with language embeddings to produce language-guided features and attention weights. Complementary static and dynamic regularization with the attention weights adaptively modulate supervision strength to prioritize semantically salient regions.

- A dynamic regularization that addresses temporal inconsistencies, effectively ensuring smooth transitions of semantic features and depth information across consecutive unit time intervals.

- A static regularization that integrates language-guided semantic and depth features into 4D Gaussians, ensuring efficient optimization while preserving semantic consistency and representational accuracy.

## 2 RELATED WORK

In this section, we provide an overview of optimization-driven methods for novel view synthesis (NVS), including approaches applicable to both dynamic and static scenes.

**Static Novel View Synthesis.**    Traditional rendering methods, including rasterization, ray tracing, path tracing, and photon mapping, simulate light-object interactions based on physical principles Li et al. (2012); Collet et al. (2015); Kanade et al. (1997); Zitnick et al. (2004). These approaches rely on geometric modelings Riegler & Koltun (2020); Zhou et al. (2018); Flynn et al. (2019); Mildenhall et al. (2019); Srinivasan et al. (2019); Thies et al. (2019); Wood et al. (2023); Kutulakos & Seitz (2000); Penner & Zhang (2017). In contrast, Neural Radiance Field (NeRF) learns a volumetric scene representation through neural networks, eliminating the need for explicit geometric models Du et al. (2021); Gao et al. (2021); Park et al. (2021a;b); Tretschk et al. (2021); Pumarola et al. (2021); Fang et al. (2022); Song et al. (2023). This enables NeRF to efficiently capture complex geometry Li et al. (2022b; 2021); Guo et al. (2023); Tian et al. (2023); Shao et al. (2023). However, NeRFs depend on volumetric rendering integration and repeated forward passes through neural networks Li et al. (2022a); Attal et al. (2023); Fridovich-Keil et al. (2023); Cao & Johnson (2023); Wang et al. (2023b); Gan et al. (2023), resulting in high training cost. 3D Gaussian Splatting (3DGS) Kerbl et al. (2023) improves efficiency by leveraging existing GPU accelerations and 3D Gaussian representations. However, these methods primarily depends on visual features for rendering and lacks direct control over semantic content Luiten et al. (2024); Li et al. (2023). L4DGS addresses this limitation by introducing a language-guided attention mechanism.

**Dynamic Novel View Synthesis.**    Traditional rendering methods convert 3D scenes into 2D images by calculating the interaction between objects and light to achieve realistic visual effects Levoy & Hanrahan (1996); Debevec et al. (1996); Gortler et al. (1996); Seitz & Dyer (1999); Buehler et al. (2001); Waechter et al. (2014). NeRF, in contrast, learns a volumetric scene representation through neural network training Mildenhall et al. (2020); Barron et al. (2021); Verbin et al. (2022);

Kopanas et al. (2022); Bemana et al. (2022), eliminating the need for complex modeling. However, NeRF's training process is time-consuming Müller et al. (2022); Yan et al. (2023); Fridovich-Keil et al. (2022); Chen et al. (2022), especially in dynamic scenes, requiring hours to days training. 3DGS Kerbl et al. (2023) improves rendering efficiency by using efficient GPU optimizations for real-time performance. However, its extensions struggle with motion blur and scene drift due to its inability to maintain temporal consistency Yang et al. (2023); Huang et al. (2024); Wu et al. (2023). L4DGS addresses these limitations by introducing dynamic and static regularization mechanisms, ensuring geometric consistency.

## 3 METHOD

We introduce Language-Guided 4D Gaussian Splatting (L4DGS), a lightweight framework that integrates natural language understanding for real-time dynamic scene synthesis using semantically-aware 4D Gaussian representations. Central to our method is a language-guided attention module that combines multi-scale CLIP-based visual features with language inputs to guide the spatial and temporal evolution of 4D Gaussians, incorporating both semantic and depth information.

### 3.1 LANGUAGE-GUIDED 4D GAUSSIAN SPLATTING

As illustrated in Fig. 1, L4DGS centers on a language-guided attention mechanism that directs the model's focus toward semantically relevant regions of the scene. To enhance visual coherence in dynamic scene rendering, we first introduce a hierarchical semantic representation that constrains 4D Gaussian distributions using object-aware features. Unlike existing approaches that rely on segmentation-based masks (e.g., subpart, part, whole) Qin et al. (2024) or external generators such as SAM Ravi et al. (2024), our method avoids explicit mask supervision. Instead, we adopt a center-differenced convolutional network Yu et al. (2020) to extract multi-scale CLIP features directly from the image, capturing fine-grained and illumination-robust semantics. To expand contextual awareness without incurring additional computational cost, we employ dilated convolutions, which increase the receptive field and support the processing of high-resolution inputs at varying feature granularities. This enriched semantic hierarchy informs the placement and refinement of 4D Gaussians, improving both the consistency and fidelity of the rendered output.

Building on this semantic foundation, L4DGS incorporates a Sparse Multi-Scale Attention (SMSA) module to adaptively fuse language input (e.g., object references or actions) with visual context (e.g., scene layout and object locations). This enables precise localization of user-referenced entities and allows for fine-grained semantic modulation during rendering. To ensure alignment between semantics and spatial structure, we introduce two complementary static regularization strategies. Static regularization enforces consistency between language-guided visual features and the rendered scene content. To further enhance the realism of spatial relationships, we leverage depth features to capture relative object positions and geometric context, especially in challenging cases involving occlusion or motion blur. Furthermore, we apply dynamic regularization to enforce temporal coherence. This term promotes smooth transitions of semantic and depth features across consecutive unit time intervals, addressing issues such as semantic drift and temporal flickering. These two mechanisms ensure that both spatial and temporal representations evolve coherently, enhancing the realism, stability, and responsiveness of dynamic scene rendering.

### 3.2 OPTIMIZATION SCHEME

**Sparse Multi-Scale Attention** To address the challenge that dynamic rendering methods lack the ability to semantically interpret or interact with the content being rendered, Sparse Multi-Scale Attention (SMSA) facilitates the interaction between language and vision, ensuring that the rendered output reflects the user's intent. To further enhance both efficiency and relevance, we introduce a top-$k$ sparse attention mechanism within SMSA that filters out irrelevant tokens and emphasizes semantically salient content. The primary function of SMSA in L4DGS is to associate language features with visual representations, directing L4DGS's focus toward spatial regions that are semantically important. By leveraging multi-scale attention mechanisms, SMSA dynamically adapts to different levels of granularity in both the language and visual inputs. This allows L4DGS to pri-

oritize regions that are contextually important, such as objects mentioned in a user's command or visually salient areas that require attention.

Furthermore, while CLIP encodes images into global semantic features, it lacks fine-grained details and struggles to accurately represent the same object across continuous time intervals. Features extracted from CLIP provide only rough boundaries for different semantic regions, leading to ambiguity and inaccuracies in 4D scene language embeddings. To learn comprehensive semantic features, we begin by extracting the language and vision features into multiple scales, then employ SMSA to compute attention at multiple levels, effectively learning the precise static scene representations. At time $t$, SMSA selects a subset of salient tokens by computing language and vision modality sparse attention. Given per-head sparse attention for modality $m \in \{v, t\}$, head $h \in \{1, \ldots, H\}$, the attention scores can be represented as $A_m^{(h,t)} = \frac{Q_m^{(h,t)} K_m^{(h,t)\top}}{\sqrt{d_h}} \in \mathbb{R}^{L_q \times L_k}$, where $Q_m$ and $K_m$ are the query and key matrices for modality $m$, respectively. To introduce sparsity, we retain only the top-$k$ key positions for each query token based on the attention scores:

$$\mathcal{I}_i^{(h,t)} = \text{TopKIndices}\big(A_m^{(h,t)}[i,:]\big), \tag{1}$$

$$\widetilde{A}_m^{(h,t)}[i,j] = \begin{cases} A_m^{(h,t)}[i,j], & \text{if } j \in \mathcal{I}_i^{(h,t)} \\ -\infty, & \text{otherwise} \end{cases}. \tag{2}$$

The sparse attention matrix $\widetilde{A}_m^{(h,t)}$ is normalized via softmax as $\bar{A}_m^{(h,t)} = \text{softmax}(\widetilde{A}_m^{(h,t)})$ and then multiplied with $V_m^{(h,t)}$ to yield $H_m^{(h,t)} = \bar{A}_m^{(h,t)} V_m^{(h,t)}$. We then concatenate and project back to obtain a visual modality feature $SMSA_v^{(t)} = \left[ H_v^{(1,t)} \| \cdots \| H_v^{(H,t)} \right] W^O$, where $SMSA_v^{(t)} \in \mathbb{R}^{L_f \times d_f}$ be the rendered features from L4DGS. By incorporating top-$k$ sparsity, SMSA dynamically attends to only the most relevant tokens across multiple spatial scales and modalities, enabling precise localization of language entities, such as the red chair. This approach not only improves the interpretability of attention but also reduces computational overhead, allowing L4DGS to operate efficiently in real-time scenarios.

**Attention-Salient Static Regularization** Central to L4DGS framework is the use of language-modulated attention weights, derived from the vision branch of the SMSA mechanism. These weights are designed to selectively emphasize spatial regions that are both semantically and structurally important, enabling L4DGS to align features in a targeted and content-aware manner. Unlike uniform regularization strategies, our method utilizes attention scores computed within SMSA to determine which visual regions warrant stronger supervision. At time $t$, we compute attention weights $w_q^{(t)} \in [0, 1]$ by first aggregating attention scores across all SMSA heads in the vision modality: $\bar{A}_v^{(t)} = \frac{1}{H} \sum_{h=1}^{H} \bar{A}_v^{(h,t)}$. We then compute the semantic importance score of each visual token $q$ by measuring the average attention it receives across all queries: $w_q^{(t)} = \frac{1}{L_f} \sum_{i=1}^{L_f} \bar{A}_v^{(t)}[i, q]$. These weights reflect how strongly the visual region $q$ is attended to under language guidance and are further normalized across valid positions: $w_q^{(t)} = \frac{w_q^{(t)}}{\sum_{j=1}^{L_f} M_j^{(t)} \cdot w_j^{(t)}}$. These language-modulated attention weights naturally highlight semantically meaningful structures within the scene, which are then used to modulate the static semantic regularization. To ensure that supervision is concentrated where it matters most for perception or interaction, language-guided static semantic regularization is defined as:

$$\mathcal{L}_{StaticSem}^* = \sum_{q=1}^{L_f} M_q^{(t)} \cdot w_q^{(t)} \left[ \| SMSA_{v,q}^{(t)} - F_{\text{rendered},q}^{(t)} \|_2^2 + \lambda \Big( 1 - \cos\big(SMSA_{v,q}^{(t)}, F_{\text{rendered},q}^{(t)}\big) \Big) \right]. \tag{3}$$

This regularization enables supervision is concentrated on regions that are semantically salient under language intent, such as object boundaries, interactable elements, and foreground structures.

Furthermore, we extend this attention mechanism to the depth domain by introducing an attention-weighted static depth regularization. The attention weights $w_q^{(t)}$, derived from the language-conditioned SMSA vision stream, are used to emphasize structurally important regions during depth supervision:

$$\mathcal{L}_{StaticDepth}^* = \sum_{q=1}^{L_f} M_q^{(t)} \cdot w_q^{(t)} \left[ \| D_{\text{rendered},q}^{(t)} - D_{GT,q}^{(t)} \|_2^2 + \lambda \Big( 1 - \cos\Big( D_{\text{rendered},q}^{(t)}, D_{GT,q}^{(t)} \Big) \Big) \right]. \tag{4}$$

| Ground Truth | Ours | Deformable4DGS | MixVoxels | HyperReel |
| --- | --- | --- | --- | --- |

Figure 2: **Qualitative Comparison on Plenoptic Video Dataset**. L4DGS outperforms leading methods in rendering hierarchical visual details, *e.g.*, the letters on a bottle inside a distant cabinet, the frequently moving faces, and the kitchenware.

This formulation enables the model to better resolve spatial relationships and occlusion patterns, focusing depth alignment on regions with higher semantic and structural relevance.

The final attention-salient static regularization is given by:

$$\mathcal{L}_{Static}^{*} = \mathcal{L}_{StaticSem}^{*} + \lambda_{Static}\mathcal{L}_{StaticDepth}^{*}. \quad (5)$$

This language-guided regularization allows L4DGS to perform semantic and geometric supervision in a spatially selective and content-aware manner.

**Lifting Representations into the 4D Space**
Ensuring temporal consistency is critical for dynamic scene rendering, especially when modeling deformable or moving objects using 4D Gaussian representations. Unlike 3D Gaussians, which capture spatial structure at a single time step, 4D Gaussians encode both spatial and temporal information. This added temporal dimension introduces challenges: semantic features associated with the same object may drift or become inconsistent across time, especially in cases of rapid motion, occlusion, or limited visual evidence. Separate-time-step supervision alone is insufficient to address these issues, as it does not constrain inter-frame coherence.

To overcome this limitation, we introduce dynamic regularization, a temporal consistency constraint that operates over continuous unit time intervals rather than isolated time steps. Specifically, we enforce smoothness in the evolution of semantic features associated with each Gaussian primitives within unit time intervals.

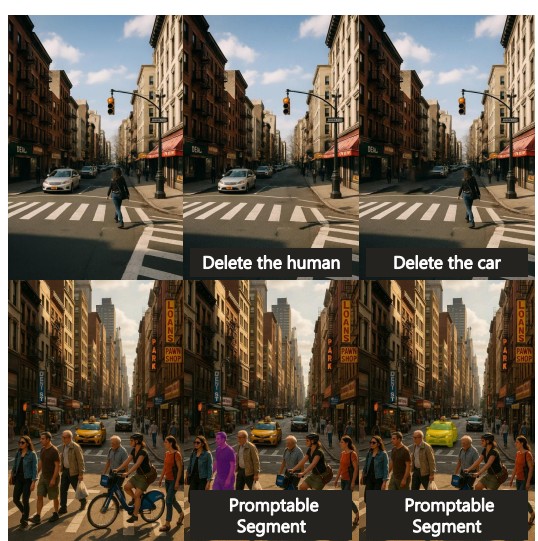

Figure 3: **Language-guided editing and promptable segmentation.**

This regularization penalizes abrupt temporal deviations, encouraging stable and coherent semantic trajectories in both spatial and temporal dimensions. It is applied directly to the learned feature embeddings, promoting continuity in appearance without modifying point cloud density.

This temporal smoothing is especially beneficial in challenging regions, such as those affected by motion blur, sparse observations, or disocclusions, where supervision from individual time steps is noisy or unreliable. By preserving the consistency of semantic distributions across unit time intervals, dynamic regularization maintains accurate object identity and geometry, reduces temporal flickering, and enhances the realism of dynamic appearance modeling. Furthermore, as this approach operates over existing Gaussian distributions, it ensures visual fidelity without increasing point cloud density.

**Dynamic Regularization** To operationalize temporal consistency across 4D Gaussians, we introduce a dynamic regularization objective that directly penalizes inconsistent feature trajectories over time. Specifically, we compute the temporal variations of both the language-guided vision features $SMSA_{v,q}^{(t)}$ and the rendered fused features $F_{rendered,q}^{(t)}$ across unit time intervals $\delta t$. For each visual token $q$, the temporal variations are defined as:

$$
\begin{aligned}
\delta SMSA_{v,q}^{(\delta t)} &= SMSA_{v,q}^{(t+\delta t)} - SMSA_{v,q}^{(t)}, \\
\delta F_{rendered,q}^{(\delta t)} &= F_{rendered,q}^{(t+\delta t)} - F_{rendered,q}^{(t)}, \\
\delta Direction &= 1 - \cos(\delta SMSA_{v,q}^{(\delta t)}, \delta F_{rendered,q}^{(\delta t)}).
\end{aligned} \tag{6}
$$

To promote semantic stability, we penalize abnormal deviations in temporal gradients through a weighted combination of feature magnitude differences and their directional misalignment:

$$
\mathcal{L}_{DynamicSem}^* = \frac{1}{L_f} \sum_{\delta t} \sum_{q=1}^{L_f} M_q^{(t)} w_q^{(t)} \left( \lambda_D \cdot \delta Direction + \|\delta SMSA_{v,q}^{(\delta t)}\|_2^2 + \|\delta F_{rendered,q}^{(\delta t)}\|_2^2 \right). \tag{7}
$$

where $M_q^{(t)} \in \{0,1\}$ masks out invalid tokens and $w_q^{(t)}$ are the attention-salient weights described previously. This loss encourages Gaussian primitives' temporal coherence in appearance, enabling smooth semantic transitions even in the presence of fast object motion, partial occlusion, or sparse frame sampling. By focusing on temporal feature gradients rather than static states, the regularization captures transferable appearance evolution, which is crucial for rendering high-frequency textures and temporally consistent reflections on dynamic surfaces.

To further ensure physically plausible motion and accurate structural evolution, we extend our formulation with a depth-based regularization that constrains temporal changes in the predicted geometry. We penalize excessive depth fluctuations in Gaussian primitives across consecutive unit time intervals:

$$
\mathcal{L}_{DynamicDepth}^* = \frac{1}{L_f} \sum_{\delta t} \sum_{q=1}^{L_f} M_q^{(t)} w_q^{(t)} \|\delta D_{rendered,q}^{(\delta t)}\|_2^2. \tag{8}
$$

where $\delta D_{rendered,q}^{(\delta t)} = D_{rendered,q}^{(t+\delta t)} - D_{rendered,q}^{(t)}$ represents the temporal change in rendered depth.

The full dynamic regularization objective integrates both semantic and geometric components:

$$
\mathcal{L}_{Dynamic}^* = \mathcal{L}_{DynamicSem}^* + \lambda_{Dynamic} \cdot \mathcal{L}_{DynamicDepth}^*. \tag{9}
$$

This dynamic regularization enhances the stability and responsiveness of L4DGS in time-varying scenes. It guides the deformation of Gaussians in a physically consistent manner, enabling the model to track non-rigid motions, handle occlusions, and interpolate missing frames in low-frame-rate or sparse input scenarios. By capturing coherent motion trajectories without increasing point cloud density, our method enables high-quality real-time rendering with reduced computational overhead.

**Semantic Consistency** Finally, to ensure consistent and semantically accurate rendering across both spatial and temporal domains, we define the comprehensive semantic consistency regularization that integrates both static and dynamic regularization components:

$$
\mathcal{L}_{4DSemantic}^* = \mathcal{L}_{Dynamic}^* + \lambda \mathcal{L}_{Static}^*. \tag{10}
$$

where $\lambda_{\mathcal{O}}$ is the learnable hyperparameter. Our approach encourages consistent representation of object semantics and depth throughout the 4D Gaussian field, effectively addressing semantic drift, flickering, and identity instability.

## 4 EXPERIMENTS

### 4.1 DATASETS AND IMPLEMENTATION DETAILS

We assess our method using two widely recognized datasets, each presenting distinct challenges in dynamic scene modeling. **Plenoptic Video Dataset Li et al. (2022b)** includes six real-world scenes, with 17 to 20 views per scene for training, and one central view reserved for evaluation. All images have a resolution of 1352×1014 Li et al. (2022b). **D-NeRF Dataset Pumarola et al. (2021)** consists of monocular video sequences from eight different scenes. Each scene contains between 50 and 200 training images, 10 to 20 validation images, and 20 test images, all resized to a resolution of

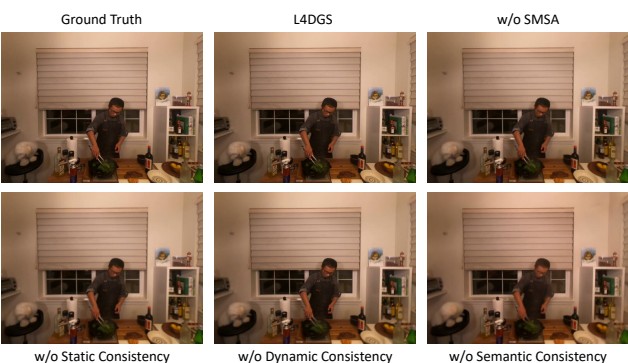

Figure 4: **Qualitative Ablation Study of Different Components in L4DGS.**

800×800 Pumarola et al. (2021). Experiments are run on a single RTX 3090 GPU. The optimization parameters are fine-tuned using the configuration settings from 3DGS Kerbl et al. (2023).

### 4.2 RESULTS

**Evaluation on Plenoptic Video Dataset.** We compare L4DGS with several state-of-the-art dynamic rendering baselines. As shown in Table 1, our method achieves the highest rendering quality by a notable margin, with a PSNR of 34.00 and an LPIPS of 0.05, outperforming leading methods in both fidelity and perceptual similarity (see Figure 2). L4DGS also demonstrates superior efficiency in training and inference: it completes training in just 30 minutes, over 3× faster than MixVoxels and more than 60× faster than K-Planes, while enabling real-time rendering at 50 FPS, exceeding existing

Table 1: **Quantitative Comparison on Plenoptic Video Dataset**. We compare L4DGS against leading methods. L4DGS obtains the highest PSNR while boosting training efficiency. *: trained on 8 GPUs and tested only on the Flame Salmon scene.

| Method | PSNR↑ | SSIM↑ | LPIPS↓ | Train↓ | FPS↑ |
|---|---|---|---|---|---|
| DyNeRF Li et al. (2022b)* | 29.58 | - | 0.08 | 1344 h | 0.015 |
| StreamRF Li et al. (2022a) | 28.16 | 0.85 | 0.31 | 79 min | 8.50 |
| HyperReel Attal et al. (2023) | 30.36 | 0.92 | 0.17 | 9 h | 2.00 |
| NeRFPlayer Song et al. (2023) | 30.69 | - | 0.11 | 6 h | 0.05 |
| K-Planes Fridovich-Keil et al. (2023) | 30.73 | 0.93 | 0.07 | 190 min | 0.10 |
| MixVoxels Wang et al. (2023b) | 30.85 | 0.96 | 0.21 | 91 min | 16.70 |
| Deformable4DGS Wu et al. (2023) | 28.42 | 0.92 | 0.17 | 72 min | 39.93 |
| Ours | 34.00 | 0.95 | 0.05 | 30 min | 50.00 |

baselines. Experimental results highlight that L4DGS not only achieves state-of-the-art visual quality but also enables substantial gains in computational efficiency, ensuring high-fidelity, real-time dynamic scene rendering with language-guided control. Figure 3 further confirms the key advantages of L4DGS. It enables language-driven control for accurate and localized scene editing.

**Evaluation on D-NeRF Dataset** We evaluate L4DGS against existing dynamic rendering methods. As shown in Table 2, L4DGS achieves the highest overall rendering quality, with a PSNR of 37.00, SSIM of 0.98, and LPIPS of 0.02. These results represent a substantial improvement over all baselines, exceeding the next-best method by over 4 dB in PSNR. Beyond accuracy, L4DGS demonstrates exceptional efficiency, requiring only 5 minutes of training, substantially faster than other real-time-capable methods such as Deformable4DGS and TiNeuVox, and enables

Table 2: **Quantitative Comparison on D-NeRF Dataset**. We compare our approach with leading dynamic scene rendering methods. L4DGS effectively balances visual quality and training efficiency in dynamic scene rendering.

| Method | PSNR↑ | SSIM↑ | LPIPS↓ | Train↓ | FPS↑ |
|---|---|---|---|---|---|
| D-NeRF Pumarola et al. (2021) | 29.17 | 0.95 | 0.07 | 24 h | 0.13 |
| TiNeuVox Fang et al. (2022) | 32.87 | 0.97 | 0.04 | 28 min | 1.60 |
| K-Planes Fridovich-Keil et al. (2023) | 31.07 | 0.97 | 0.02 | 54 min | 1.20 |
| FFDNeRF Guo et al. (2023) | 31.70 | 0.96 | 0.05 | - | <1.20 |
| MSTH Wang et al. (2023a) | 30.40 | 0.97 | 0.05 | 9.80 min | - |
| V4D Gan et al. (2023) | 32.67 | 0.97 | 0.05 | 10.21 h | 2.64 |
| Deformable4DGS Wu et al. (2023) | 32.99 | 0.97 | 0.05 | 13 min | 104.00 |
| Ours | 37.00 | 0.98 | 0.02 | 5 min | 150.00 |

real-time rendering at 150 FPS. These findings confirm the effectiveness of L4DGS in jointly ensuring high visual fidelity, rapid training, and real-time performance.

Table 3: **Ablation Study with Quantitative Comparison on D-NeRF Dataset**. We validate different components in L4DGS on rendering quality PSNR.

| ID | Dyn. Sem. | Stat. Sem. | Dyn. Depth | Stat. Depth | Attn. Focus | Jumping Jacks | Mutant | Stand Up |
|---|---|---|---|---|---|---|---|---|
| a |  |  |  |  |  | 34.33 | 35.87 | 35.91 |
| b | ✓ |  |  |  |  | 35.85 | 37.76 | 38.03 |
| c |  | ✓ |  |  |  | 35.47 | 37.29 | 37.50 |
| d |  |  | ✓ |  |  | 35.09 | 36.81 | 36.97 |
| e |  |  |  | ✓ |  | 34.71 | 36.34 | 36.44 |
| f | ✓ | ✓ | ✓ | ✓ |  | 37.96 | 40.36 | 41.14 |
| Full | ✓ | ✓ | ✓ | ✓ | ✓ | 38.44 | 40.89 | 41.50 |

## 4.3 ABLATION STUDIES

**Language-Guided Semantics Consistency.** To evaluate the impact of language-guided semantic consistency, we conduct an ablation study that retains only the static semantic regularization module. As shown in Table 3 (c), this configuration consistently outperforms the non-regularized baseline (Table 3 (a)), highlighting the importance of aligning rendered content with language-conditioned visual features. The results demonstrate that language-guided spatial attention serves as strong supervision for rendering salient structures and object boundaries, which is a crucial foundation for achieving spatiotemporal consistency.

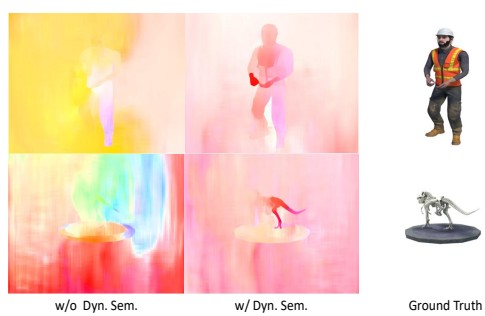

w/o Dyn. Sem.   w/ Dyn. Sem.   Ground Truth

Figure 5: **Optical Flow Visualization.**

Furthermore, to evaluate the effect of attention-salient weighting on regularization, we compare the full model (Table 3 *Full*) with its ablated variant (Table 3 (*f*)). The inclusion of the attention-salient module yields consistent improvements across all scenes, with especially pronounced gains in detail-rich scenes such as Mutant. These results indicate that attention-aware weighting enhances the effectiveness and spatial selectivity of supervision, improving cross-modal alignment and rendering quality in structurally complex regions.

**Dynamic Consistency.** To isolate the impact of temporal semantic consistency, we evaluate a model variant (Table 3 (*b*)), which shows a notable performance gain over the baseline (Table 3 (*a*)). This demonstrates the effectiveness of enforcing temporal coherence in semantic space. Among all components, dynamic semantic supervision yields the highest average PSNR improvement across scenes, underscoring its central role in addressing temporal flickering and semantic drift. In Figure 4, qualitative results further reveal that omitting dynamic regularization introduces artifacts such as motion blur. These findings confirm that temporal feature alignment is essential for robust, high-fidelity dynamic scene rendering.

## 5 CONCLUSION

We have presented L4DGS, a language-guided framework for real-time dynamic scene rendering based on semantically enriched 4D Gaussian representations. By integrating natural language understanding with hierarchical visual features through a Sparse Multi-Scale Attention (SMSA) mechanism, our approach enables language-guided, fine-grained rendering in complex dynamic environments. To ensure spatial and temporal consistency, we introduce static and dynamic regularization strategies that align semantic and depth features across both space and time, effectively addressing temporal semantic drift and inconsistency. Extensive experiments demonstrate that incorporating language semantics into the rendering pipeline substantially ensures realistic rendering and enhances scene interpretability, while maintaining comparable training efficiency.

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
