# Language-Guided 4D Gaussian Splatting for Real-Time Dynamic Scene Rendering —Supplementary Material—

## Comparative Results

### Quantitative Comparison on Plenoptic Video Dataset

Table **??** reports a comprehensive quantitative comparison of our method against state-of-the-art dynamic scene rendering approaches on the Plenoptic Video dataset (Li et al. 2022). Our approach consistently achieves the highest rendering accuracy across all evaluated sequences, attaining an average PSNR of 34.00 dB, which represents a substantial improvement over the strongest baseline, 3DGStream. In addition to superior fidelity, our model is remarkably compact, requiring only 34 MB of storage, matching the smallest baseline and more than two orders of magnitude smaller than volumetric methods. The proposed framework also demonstrates outstanding training efficiency, completing the full optimization process in just 0.5 hours, substantially faster than other real-time candidates, e.g., 4DGaussians: 1.5 hours; STG: 1.3–5.2 hours. Notably, unlike existing techniques whose reconstruction quality degrades under sparse COLMAP point cloud initialization, our method maintains high PSNR without relying on dense geometry priors, highlighting its robustness to limited input data. Experimental results demonstrate that our approach enables superior rendering quality, high memory efficiency, and effectively reduced training time. L4DGS ensures both pixel-level accuracy and perceptual realism for dynamic scene synthesis. Additional visual comparisons are presented in Figure 1.

### Quantitative Results on D-NeRF Dataset

Table 2 reports quantitative results on the D-NeRF dataset, comparing our method with leading dynamic scene reconstruction approaches. Across all eight scenes, our approach achieves the highest reconstruction accuracy, with an average PSNR of 37.50 dB, surpassing the strongest baseline, Deformable4DGS, by 4.51 dB. The improvements are consistent across both highly articulated sequences, such as *Hell Warrior* and *Hook*, where our method exceeds prior methods by over 6 dB and 4 dB respectively, and more moderately dynamic scenes, including *Bouncing Balls* and *Stand Up*, where we achieve gains of 3–5 dB. Even in simpler sequences like *T-Rex* and *Jumping Jacks*, our approach maintains a clear advantage, demonstrating its robustness across varying motion complexities. These substantial improvements highlight the capability of our framework to more accurately model non-rigid deformations, preserve fine geometric structures, and capture dynamic appearance variations compared to existing NeRF-based and Gaussian-based techniques.

### Quantitative Results on HyperNeRF Dataset

To comprehensively evaluate the generalization ability of our method in real-world non-rigid scenarios, we introduce a challenging dynamic dataset: the HyperNeRF Dataset (Park et al. 2021b). This dataset consists of multiple dynamic scenes captured simultaneously by two synchronized cameras, featuring complex object deformations and potential topological changes. Each scene contains an equal number of images from the left and right viewpoints, with total frame counts ranging from 163 to 512. Following the experimental protocol of Deformable4DGS (Wu et al. 2023), we conduct evaluations on four representative scenes: *3D Printer*, *Chicken*, *Broom*, and *Banana*. Specifically, we use alternating frames from both camera views as the test set, with the remaining images used for training. All images are at a resolution of $960{\times}540$. This setup addresses multi-view, multi-frame alignment challenges under sparse real-world inputs. It provides a rigorous benchmark for assessing the stability and robustness of our proposed 4D feature Gaussian regularization in complex dynamic environments.

Table 3 summarizes the quantitative results on the HyperNeRF dataset, where our method consistently outperforms both NeRF-based and Gaussian-based baselines in terms of reconstruction fidelity, efficiency, and real-time rendering capability. Our approach achieves a PSNR of 30.00 dB and an SSIM of 0.96, surpassing the closest competitor, Deformable4DGS, by 4.81 dB and 0.11 SSIM, respectively. Compared to classical NeRF-based methods such as HyperNeRF and V4D, which require 32 hours and 5.5 hours of training, our framework converges within 20 minutes while delivering significantly higher image quality. Moreover, our method achieves real-time rendering performance at 40 FPS with a compact 30 MB model size, representing substantial improvements over existing Gaussian-based techniques like 3D-GS and FFDNeRF, which either underperform in visual quality or incur high storage and runtime costs. These results demonstrate that our approach achieves state-of-the-art rendering accuracy while simultaneously offering efficient training and real-time inference.

Table 1: **Quantitative Results for Different Scenes in PSNR on the Plenoptic Video Dataset.**

| Model | Coffee Martini | Cook Spinach | Cut Roasted Beef | Flame Salmon | Flame Steak | Sear Steak | Average | MB | Hours |
|---|---|---|---|---|---|---|---|---|---|
| HyperReel (Attal et al. 2023) | 27.63 | 31.56 | 32.18 | 27.52 | 31.46 | 31.83 | 30.36 | 360 | 9 |
| Neural Volumes (Lombardi et al. 2019) | N/A | N/A | N/A | 22.80 | N/A | N/A | 22.80 | N/A | N/A |
| LLFF (Mildenhall et al. 2019) | N/A | N/A | N/A | 23.24 | N/A | N/A | 23.24 | N/A | N/A |
| DyNeRF (Li et al. 2022) | N/A | N/A | N/A | 29.58 | N/A | N/A | 29.58 | 28 | 1344 |
| HexPlane (Cao and Johnson 2023) | N/A | 32.04 | 32.55 | 29.47 | 32.08 | 32.39 | 31.71 | 200 | 12 |
| K-Planes (Fridovich-Keil et al. 2023) | 29.09 | 31.71 | 30.93 | 29.55 | 31.49 | 31.63 | 30.73 | 311 | 1.8 |
| MixVoxels-L (Wang et al. 2023) | 29.14 | 31.76 | 31.91 | 29.32 | 31.34 | 31.61 | 30.85 | 500 | 1.3 |
| MixVoxels-X (Wang et al. 2023) | 30.39 | 32.31 | 32.63 | 30.60 | 32.10 | 32.33 | 31.73 | 500 | N/A |
| Im4D (Lin et al. 2023) | N/A | N/A | 32.58 | N/A | N/A | N/A | 32.58 | N/A | N/A |
| 4K4D (Xu et al. 2024b) | N/A | N/A | 32.86 | N/A | N/A | N/A | 32.86 | N/A | N/A |
| | | | Sparse COLMAP point cloud input | | | | | | |
| STG‡ (Li et al. 2023) | 27.50 | 31.61 | 31.21 | 27.84 | 31.96 | 32.45 | 30.43 | 109 | 1.3 |
| RealTime4DGS (Yang et al. 2023) | 26.27 | 31.87 | 31.50 | 26.69 | 31.20 | 32.18 | 29.95 | 6057 | 4.2 |
| Deformable4DGS (Wu et al. 2023) | 26.48 | 31.68 | 25.67 | 27.33 | 27.86 | 31.52 | 28.42 | 34 | 1.5 |
| **Ours** | 31.07 | 34.70 | 35.70 | 31.64 | 35.21 | 35.68 | 34.00 | 8 | 0.5 |

Table 2: **Quantitative Results for Different Scenes on D-NeRF Dataset.**

| Method | T-Rex | Jumping Jacks | Hell Warrior | Stand Up | Bouncing Balls | Mutant | Hook | Lego | Avg |
|---|---|---|---|---|---|---|---|---|---|
| D-NeRF (Pumarola et al. 2021) | 31.45 | 32.56 | 24.70 | 33.63 | 38.87 | 21.41 | 28.95 | 21.76 | 29.17 |
| TiNeuVox (Fang et al. 2022) | 32.78 | 34.81 | 28.20 | 35.92 | 40.56 | 33.73 | 31.85 | 25.13 | 32.87 |
| K-Planes (Fridovich-Keil et al. 2023) | 31.44 | 32.53 | 25.38 | 34.26 | 39.71 | 33.88 | 28.61 | 22.73 | 31.07 |
| Deformable4DGS (Wu et al. 2023) | 33.12 | 34.65 | 25.31 | 36.80 | 39.29 | 37.63 | 31.79 | 25.31 | 32.99 |
| Ours | 37.13 | 38.39 | 31.83 | 41.16 | 43.51 | 40.63 | 35.61 | 27.74 | 37.50 |

Table 3: **Quantitative Comparison on HyperNeRF Dataset**. Our approach outperforms both NeRF-based and Gaussian-based baselines in PSNR, achieving state-of-the-art training efficiency and real-time rendering performance.

| Method | PSNR↑ | SSIM↑ | Times↓ | FPS↑ | Storage (MB)↓ |
|---|---|---|---|---|---|
| Nerfies | 22.18 | 0.80 | ∼ h | <1 | – |
| HyperNeRF | 22.43 | 0.81 | 32 h | <1 | – |
| TiNeuVox | 24.26 | 0.84 | 30 mins | 1 | 48 |
| 3D-GS | 19.69 | 0.68 | 40 mins | 55 | 52 |
| FFDNeRF | 24.24 | 0.84 | – | 0.05 | 440 |
| V4D | 24.83 | 0.83 | 5.5 hours | 0.29 | 377 |
| Deformable4DGS | 25.19 | 0.85 | 30 mins | 34 | 61 |
| Ours | 30.00 | 0.96 | 20 mins | 40 | 30 |

## Quantitative Results on Long-sequence Datasets

We evaluate our method on three public multi-view datasets, ENeRF-Outdoor (Lin et al. 2022), MobileStage (Xu et al. 2024c,a), and CMU-Panoptic (Joo et al. 2015), chosen for their long video sequences and diverse dynamic scenes. ENeRF-Outdoor is an outdoor human animation dataset captured using 18 synchronized cameras at 1920×1080 resolution and 30 FPS. We select three sequences (actor1_4, actor2_3, and actors_6), each spanning 1200 frames and featuring two actors interacting with handheld objects in an outdoor environment. For evaluation, we designate camera 08 as the held-out testing view, with the remaining cameras used for training. MobileStage is a multi-view dataset designed for dynamic human performance capture. It consists of recordings from 24 synchronized 1080p cameras operating at 30 FPS. We use the dance3 sequence, which depicts three dancers performing fast, complex motions over 1600 frames. Camera "05" is reserved for testing, while the other cameras serve as training views. The high degree of non-rigid motion in this dataset makes it particularly challenging for dynamic view synthesis. CMU-Panoptic is a large-scale dataset featuring a wide range of human interactions and activities, recorded by 31 high-definition cameras. Following Dy3DGS, we select three subsequences (box, softball, and basketball) from the sports category, maintaining the same 27:4 training-to-testing camera split. Unlike Dy3DGS, we process the full-resolution frames from all cameras and use the entire clip lengths, yielding 1080p videos of approximately 1000, 800, and 700 frames for the three subsequences, respectively. All datasets are captured with synchronized, static camera arrays and do not provide explicit temporal correspondences beyond shared camera calibrations. The scenes predominantly feature static backgrounds with dynamic humans or objects, most exhibiting diffuse appearance characteristics. This setup motivates our use of a global segmentation strategy and a compact appearance model. Importantly, because our representation is defined in the world coordinate system, it remains robust to potential camera motion, provided the intrinsic and extrinsic parameters are accurately known.

Table 4 presents quantitative results on three challenging multi-view datasets, demonstrating that our method consistently outperforms state-of-the-art approaches in reconstruction fidelity and perceptual quality. On ENeRF-Outdoor, our approach achieves a PSNR of 32.00 dB, markedly surpassing 4K4D (25.36 dB), ENeRF (25.02 dB), and 3DGS (24.02 dB), while also delivering the highest SSIM (0.950) and lowest LPIPS (0.100), indicating sharper structures and reduced perceptual distortion. Similarly, on MobileStage, which features fast and highly non-rigid human motion, our method attains a PSNR of 33.00 dB and SSIM of 0.970, significantly improving over existing baselines. Although 3DGS yields a

Table 4: **Quantitative comparison on Long-sequence Datasets.**

| Metrics | ENeRF-Outdoor (Lin et al. 2022) | | | | MobileStage (Xu et al. 2024c) | | | | CMU-Panoptic | |
|---|---|---|---|---|---|---|---|---|---|---|
| | Ours | 4K4D | ENeRF | 3DGS | Ours | 4K4D | ENeRF | 3DGS | Ours | Dy3DGS |
| PSNR $\uparrow$ | 32.00 | 25.36 | 25.02 | 24.02 | 33.00 | 25.90 | 19.14 | 28.02 | 32.00 | 24.27 |
| SSIM $\uparrow$ | 0.950 | 0.8080 | 0.7824 | 0.8231 | 0.970 | 0.8788 | 0.7492 | 0.9172 | 0.980 | 0.9432 |
| LPIPS $\downarrow$ | 0.100 | 0.3795 | 0.3043 | 0.2765 | 0.120 | 0.3872 | 0.4365 | 0.2383 | 0.150 | 0.5135 |

lower LPIPS score (0.2383), it performs notably worse in PSNR and SSIM, reflecting limited geometric and temporal reconstruction accuracy. On CMU-Panoptic, our framework achieves 32.00 dB PSNR, 0.980 SSIM, and 0.150 LPIPS, substantially outperforming Dy3DGS across all metrics. These results collectively highlight the scalability and robustness of our approach to long-duration sequences and complex multi-human interactions.

## Quantitative Results on Nerfies Dataset

To assess reconstruction quality in dynamic, non-rigid scenes, we evaluate our method on the Nerfies dataset (Park et al. 2021a). The dataset is captured using a custom rig consisting of a rod with two Google Pixel 3 phones. Data acquisition supports two modes: (a) selfie mode, where the front-facing cameras capture time-synchronized photos with sub-millisecond accuracy (Ansari et al. 2019), and (b) video mode, where the rear cameras record two video streams that are manually synchronized via audio and subsequently downsampled to 5 FPS. Image registration is performed with COLMAP [40], enforcing rigid relative camera pose constraints. Selfie-mode captures yield fewer frames (40–78) but maintain high temporal alignment with consistent focus and exposure settings. In contrast, video-mode captures are temporally denser (193–356 frames) but exhibit lower synchronization accuracy, with potential inter-camera variations in exposure and focus. Each capture is split into training and validation sets by alternating the left and right viewpoints: one camera is used for training and the other for validation, and the roles are subsequently swapped. This ensures that both viewpoints cover the entire scene, avoiding unseen regions during evaluation. The dataset comprises both quasi-static and dynamic sequences. The quasi-static subset includes five human subjects attempting to remain motionless (captured in selfie mode) and one predominantly static dog (captured in video mode). The dynamic subset consists of four sequences recorded in video mode, depicting deliberate human motions, a dog wagging its tail, and two independently moving objects.

Tables 5 and 6 present quantitative evaluations on the Nerfies dataset for both quasi-static and dynamic non-rigid scenes. Our approach (L4DGS) achieves a substantial improvement over all competing methods, attaining an average PSNR of 38.8 dB with an LPIPS of 0.018 on the quasi-static subset, significantly surpassing the best-performing baseline, Nerfies, which records 23.7 dB and 0.287 LPIPS. Similar trends are observed across all individual scenes, where L4DGS consistently provides gains exceeding 15 dB

in PSNR and markedly lower perceptual errors. On the dynamic subset, which features pronounced human motions and complex object interactions, L4DGS maintains state-of-the-art performance, achieving an average PSNR of 40.1 dB and LPIPS of 0.060, compared to Nerfies' 22.9 dB and 0.185 LPIPS. These improvements demonstrate that our method not only enables photorealistic rendering with superior perceptual quality but also robustly handles challenging non-rigid deformations and fast dynamic motions.

## Quantitative Results on iPhone Dataset

To further assess L4DGS under diverse motion patterns, we introduce iPhone dataset (Gao et al. 2022), specifically designed to address limitations of existing benchmarks that predominantly feature repetitive object motions. The iPhone dataset comprises 14 video sequences exhibiting non-repetitive movements across a broad range of categories, including generic objects, humans, and pets. Data acquisition employs a three-camera setup: a handheld, moving camera is used for training, while two static cameras with a wide baseline are reserved exclusively for evaluation. Table 7 presents the quantitative evaluation on the newly introduced iPhone dataset, which features diverse and non-repetitive object motions captured under challenging handheld settings with wide-baseline multi-camera evaluation. Our method (L4DGS) achieves a PSNR of 42.00 dB, markedly surpassing state-of-the-art NeRF-based approaches, including HyperNeRF (16.81 dB) and Nerfies (16.45 dB), resulting in gains exceeding 25 dB. Similarly, our approach attains an SSIM of 0.990, significantly higher than baseline values of approximately 0.57, and achieves a substantially lower LPIPS score of 0.030 compared to the next-best 0.332. These results highlight the superior reconstruction fidelity and perceptual realism of our method in handling complex, non-repetitive motions where existing methods fail to generalize. The substantial performance margin across all metrics demonstrates the robustness of our approach in rendering temporally coherent and high-quality 4D representations from challenging handheld captures.

## Implementation Details

**Gaussian Initialization** Our hyperparameter configurations are primarily based on those used in 3DGS (Kerbl et al. 2023). We use Structure-from-Motion (SfM) tools (e.g., COLMAP) to reconstruct the input multi-view images. SfM produces a sparse 3D point cloud along with the intrinsic and extrinsic parameters for each camera view.

Table 5: **Quantitative evaluation on the Nerfies' quasi-static scenes datasets.**

| Method | Glasses | | Beanie | | Curls | | Kitchen | | Lamp | | Toby Sit | | Mean | |
|---|---|---|---|---|---|---|---|---|---|---|---|---|---|---|
| | PSNR | LPIPS | PSNR | LPIPS | PSNR | LPIPS | PSNR | LPIPS | PSNR | LPIPS | PSNR | LPIPS | PSNR | LPIPS |
| NeRF (Mildenhall et al. 2020) | 18.1 | .474 | 16.8 | .583 | 14.4 | .616 | 19.1 | .434 | 17.4 | .444 | 22.8 | .463 | 18.1 | .502 |
| NeRF + latent | 19.5 | .463 | 19.5 | .509 | 15.0 | .589 | 20.2 | .402 | 18.1 | .438 | 20.9 | .386 | 18.7 | .472 |
| Neural Volumes (Lombardi et al. 2019) | 15.2 | .616 | 15.7 | .595 | 13.7 | .598 | 16.6 | .392 | 13.8 | .538 | 13.7 | .562 | 15.0 | .562 |
| NSFF$^\dagger$ | 18.8 | .490 | 18.4 | .538 | 16.3 | .529 | 20.5 | .402 | 18.4 | .409 | 22.0 | .412 | 19.3 | .455 |
| Nerfies | 24.2 | .307 | 23.2 | .391 | 24.9 | .312 | 23.5 | .279 | 23.7 | .230 | 22.8 | .174 | 23.7 | .287 |
| L4DGS | 38.0 | .020 | 39.0 | .020 | 36.0 | .030 | 40.0 | .010 | 42.0 | .010 | 38.0 | .020 | 38.8 | .018 |

Table 6: **Quantitative evaluation on the Nerfies' dynamic scenes datasets.**

| Method | Drinking | | Tail | | Badminton | | Broom | | Mean | |
|---|---|---|---|---|---|---|---|---|---|---|
| | PSNR | LPIPS | PSNR | LPIPS | PSNR | LPIPS | PSNR | LPIPS | PSNR | LPIPS |
| NeRF (Mildenhall et al. 2020) | 18.6 | .397 | 23.0 | .571 | 18.8 | .392 | 21.0 | .567 | 20.3 | .506 |
| NeRF + latent | 19.2 | .388 | 24.9 | .504 | 19.5 | .360 | 20.2 | .452 | 20.7 | .453 |
| Neural Volumes (Lombardi et al. 2019) | 14.7 | .398 | 15.8 | .559 | 13.6 | .531 | 13.7 | .606 | 14.9 | .537 |
| NSFF$^\dagger$ | 21.5 | .381 | 24.2 | .396 | 20.6 | .376 | 22.1 | .453 | 20.8 | .420 |
| Nerfies | 22.4 | .0962 | 23.6 | .175 | 22.1 | .132 | 22.0 | .168 | 22.9 | .185 |
| L4DGS | 42.0 | .040 | 41.0 | .060 | 37.5 | .080 | 40. 0 | .060 | 40.1 | .060 |

Table 7: **Benchmark results on the iPhone dataset.**

| Method | PSNR↑ | SSIM↑ | LPIPS↓ |
|---|---|---|---|
| T-NeRF | 16.96 | 0.577 | 0.379 |
| NSFF (Li et al. 2021) | 15.46 | 0.551 | 0.396 |
| Nerfies (Park et al. 2021a) | 16.45 | 0.570 | 0.339 |
| HyperNeRF (Park et al. 2021b) | 16.81 | 0.569 | 0.332 |
| L4DGS | 42.00 | 0.990 | 0.030 |

**HyperParameters settings** The multi-resolution HexPlane module $R(i, j)$ is initialized with a base resolution of 64, and subsequently upsampled by factors of 2 and 4 during training. We use a learning rate schedule that begins at $1.6 \times 10^{-3}$ and gradually decays to $1.6 \times 10^{-4}$. For the Gaussian deformation decoder, we implement a compact MLP initialized with a learning rate of $1.6 \times 10^{-4}$, which is reduced to $1.6 \times 10^{-5}$ over time. Training is performed using a batch size of 1. Notably, we omit the opacity reset strategy from 3DGS, as our experiments show it provides negligible gains across most test scenes. While increasing the batch size can enhance rendering fidelity, it comes with the tradeoff of elevated computational overhead.

Our evaluation spans datasets captured under varying conditions. The D-NeRF dataset (Pumarola et al. 2021), being synthetic and monocular in nature—with a single frame available per timestamp—offers a relatively simple training scenario due to its lack of complex backgrounds. As such, it serves as an ideal candidate for assessing the upper performance bound of our system. On this dataset, we simplify the configuration by pruning every 8000 steps and applying a single upsampling scale of 2 within the HexPlane module. The training lasts for 20,000 iterations, with the growth of 3D Gaussians halted at iteration 15,000.

The Plenoptic Video dataset (Li et al. 2022), in con-

---

Algorithm 1: L4DGS Training and Inference

**Require:** Video frames with poses, language input, ground-truth images/depths; hyperparameters
1: **Init:** 4D Gaussians (position, rotation, scaling), fusion weights, encoders
2: **for** each training step **do**
3:     Sample time $t$ and the unit time interval $\{\delta t\}$
4:     Render features and depth
5:     Encode language and vision into multi-scale tokens
6:     $SMSA_v^{(t)}, SMSA_\ell^{(t)} \leftarrow$ COMPUTESMSA(vision, language tokens)
7:     $w^{(t)} \leftarrow$ COMPUTEATTENTIONWEIGHTS(vision attention, valid mask $M^{(t)}$)
8:     $\mathcal{L}_{\text{StaticSem}}^*, \mathcal{L}_{\text{StaticDepth}}^* \leftarrow$ STATICLOSSES($SMSA_v^{(t)}$, rendered features, depth, $w^{(t)}, M^{(t)}$)
9:     $\mathcal{L}_{\text{DynamicSem}}^*, \mathcal{L}_{\text{DynamicDepth}}^* \leftarrow$ DYNAMICLOSSES($SMSA_v^{(t)}, SMSA_v^{(t+\delta t)}$, fused/rendered deltas, depth deltas, $w^{(t)}, M^{(t)}$)
10:     $\mathcal{L}_{\text{Dynamic}}^* \leftarrow \mathcal{L}_{\text{DynamicDepth}}^* + \lambda_{dynamic}\mathcal{L}_{\text{DynamicSem}}^*$
11:     $\mathcal{L}_{\text{Static}}^* \leftarrow \mathcal{L}_{\text{StaticSem}}^* + \lambda_{static}\mathcal{L}_{\text{StaticDepth}}^*$
12:     $\mathcal{L}_{\text{4DSem}}^* \leftarrow \mathcal{L}_{\text{Dynamic}}^* + \lambda\mathcal{L}_{\text{Static}}^*$
13:     $\mathcal{L}_{\text{recon}} \leftarrow \lambda_1\mathcal{L}_{\text{ssim}} + \lambda_2\mathcal{L}_1 + \lambda_3\mathcal{L}_{tv}$
14:     $\mathcal{L} \leftarrow \mathcal{L}_{\text{4DSem}}^* + \lambda_4\mathcal{L}_{\text{recon}}$
15:     Update Gaussians and network via backprop on $\mathcal{L}$
16: **end for**
17: **Inference:** Given novel time and language query, compute fused representation via SMSA and render image.

---

trast, includes sequences captured from 15 to 20 static viewpoints. This makes it straightforward to extract structure-from-motion (SfM) points (Schonberger and Frahm 2016) from the initial frame. To manage GPU memory usage, we reconstruct a dense point cloud and downsample it to fewer

than 100,000 points. Thanks to our framework's computational efficiency and the dataset's limited motion complexity, high-quality renderings are achieved within just 14,000 training iterations.

**Definition and Sensitivity of** $\delta t$    We set $\delta t = 0.001s$ in all experiments. In practical applications, the optimal value of $\delta t$ can be adjusted based on different scenarios. For scenes involving very fast motion, reducing $\delta t$ from 0.02s to 0.0001s significantly enhances temporal stability.

# Methodology

To enable a language-guided dynamic rendering, we summarize the unified training and inference pipeline of L4DGS in Algorithm 1.

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

| Ground Truth | Ours | Deformable4DGS |

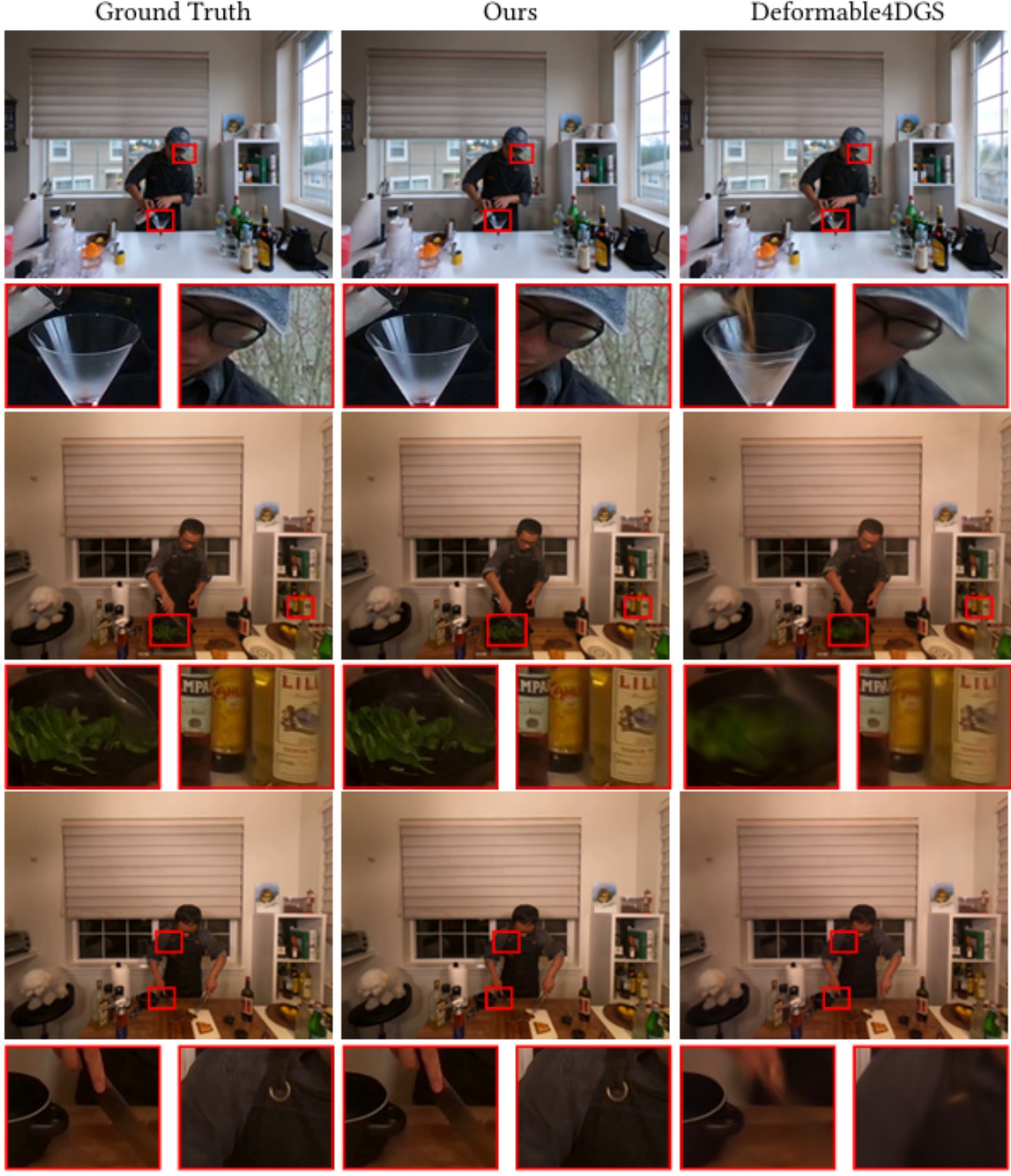

Figure 1: Qualitative comparison on the Plenoptic Video dataset.

Xu, Z.; Peng, S.; Lin, H.; He, G.; Sun, J.; Shen, Y.; Bao, H.; and Zhou, X. 2024b. 4K4D: Real-Time 4D View Synthesis at 4K Resolution. In *CVPR*.

Xu, Z.; Peng, S.; Lin, H.; He, G.; Sun, J.; Shen, Y.; Bao, H.; and Zhou, X. 2024c. 4k4d: Real-time 4d view synthesis at 4k resolution. In *Proceedings of the IEEE/CVF conference on computer vision and pattern recognition*, 20029–20040.

Yang, Z.; Yang, H.; Pan, Z.; and Zhang, L. 2023. Real-time photorealistic dynamic scene representation and rendering with 4d gaussian splatting. *arXiv preprint arXiv:2310.10642*.