# OpenReview forum: "Language-Guided 4D Gaussian Splatting for Real-Time Dynamic Scene Rendering"
_ICLR.cc/2026/Conference — Submitted to ICLR 2026_

### Official Review · Reviewer_aJtV · 2025-10-31

**Soundness:** 2
**Presentation:** 2
**Contribution:** 3
**Rating:** 4
**Confidence:** 4

**Summary:**

This paper introduces a language-guided regularization loss to enhance semantic consistency and rendering quality in 4D Gaussian Splatting (4DGS).

**Strengths:**

The idea of integrating language guidance into 4DGS is interesting. The reported metric is high.

**Weaknesses:**

1. Unclear relevance of physical motion to language guidance

 (line 354 to line 369) "To further ensure physically plausible motion and accurate structural evolution, we extend our formulation with a depth-based regularization that constrains temporal changes in the predicted geometry. We penalize excessive depth fluctuations in Gaussian primitives across consecutive unit time intervals:" The connection between motion regularization and language semantics is not well explained.


2. Inadequate evaluation of “Language-Guided Semantic Consistency”

The semantic consistency claim is mainly demonstrated through optical flow visualizations. This seems inappropriate because optical flow operates at the pixel level and does not directly measure semantic or language-level consistency. More suitable semantic metrics and visualziation should be considered to substantiate this claim.

3. lack of context for Figure 3
The paper states that “Figure 3 further confirms the key advantages of L4DGS. It enables language-driven control for accurate and localized scene editing.” However, Figure 3 is not clearly connected to 4DGS or explained in sufficient context. The figure and its caption should better demonstrate how language guidance enables localized editing within the 4DGS framework.

4. Missing analysis of computational overhead
The paper lacks an analysis of the computational cost introduced by additional regularization terms (e.g., depth-based and CLIP-based losses). It would be helpful to quantify the impact on training speed and memory usage compared to vanilla 4DGS.

**Questions:**

1. Can the authors provide concrete examples or quantitative results showing “language-driven control for accurate and localized scene editing”? Ideally, please show examples on datasets of 4DGS

2. Training details and runtime feasibility
The paper claims that all experiments were conducted on a single RTX 3090 GPU and cost 20~30 minutes. How was this runtime measured?
Is the CLIP-based language regularization computed online during training for all 20,000 or 14000 iterations?
What image resolution is used as input to the CLIP model during training?

---

> ### Author Response · Authors · 2025-11-16
> **Response to Reviewer aJtV**
>
> W1: This sentence is not emphasizing the relationship between physical motion and language guidance; rather, it highlights the benefit of extending the constraints on physical motion and language guidance to a continuous unit time interval.
>
> W2: Semantic consistency manifests as stable rendering quality in dynamic scenes. Optical-flow consistency further demonstrates the coherence and stability of L4DGS when rendering motion in such scenes.
>
> W3 & Q1: L4DGS is a real-time dynamic rendering method that enables language-guided image editing. As shown in Fig. 3, when performing deletion or segmentation, L4DGS can accurately localize the corresponding objects in the scene and apply the requested edits.
>
> W4 & Q2: In Tables 1 and 2, we provide a detailed comparison of the training time and FPS of different methods. The performance gains of L4DGS stem directly from the incorporation of semantic consistency regularization. This design improves not only rendering quality but also overall efficiency, including faster training. We evaluate all baselines using a unified testing benchmark, following the protocol of 4DRotorGS [2]. The language regularization term measures the discrepancy between rendered features and ground truth, and its loss is computed only after rendering is completed. Training time is mainly constrained by hardware and network complexity. Regularization adds minimal overhead. Once bandwidth limits are reached, we observe substantial accuracy gains without loss of speed.
>
> [2] Duan, Yuanxing, et al. "4d-rotor gaussian splatting: towards efficient novel view synthesis for dynamic scenes." ACM SIGGRAPH 2024 Conference Papers. 2024.

---

### Official Review · Reviewer_ZLKE · 2025-11-01

**Soundness:** 3
**Presentation:** 3
**Contribution:** 2
**Rating:** 4
**Confidence:** 4

**Summary:**

This paper introduces language-guided 4D Gaussian Splatting (L4DGS), a lightweight framework that integrates natural language guidance into real-time 4D Gaussian splatting for dynamic scene rendering. The paper points out that although the existing 3DGS has improved the rendering efficiency, it still lacks semantic control capabilities. The dynamic scene expansion version also has problems such as motion blur and scene drift. For the above issues, this paper combines the sparse multi-scale Attention (SMSA) used for cross-modal feature fusion with static and dynamic regularization mechanisms to ensure semantic and temporal consistency. It significantly outperforms existing methods in terms of rendering fidelity and computational efficiency, and the qualitative results demonstrate its ability to guide scene editing through language (such as deleting target objects).

**Strengths:**

1. The current mainstream dynamic rendering methods mostly rely on visual features and lack explicit control at the semantic level, making it impossible to align human language intentions with dynamic rendering results. L4DGS innovatively constructs a language-guided 4DGS framework, deeply integrating the 4DGS representation of natural language understanding and semantic perception.
2. The proposed Sparse multi-scale Attention (SMSA) effectively aligns language and visual patterns and uses the top-k sparse strategy to improve interpretability and efficiency. The dual regularization design (static + dynamic) ensures spatial consistency and temporal consistency, and resolves the long-standing semantic drift and flickering issues in dynamic NeRF systems.
3. Compared with leading baselines such as MixVoxels, K-Planes and Deformable4DGS, there have been substantial improvements in PSNR, LPIPS and rendering speed. The training time is reduced to a few minutes while maintaining real-time rendering quality. Support for prompt semantic operations (" delete car ", "delete person") highlights the potential of interactive applications (for example, VR/AR, robot perception, content creation).

**Weaknesses:**

1. Although the paper emphasizes "first language-embedded real-time 4D rendering", there have already been many works combining 4DGS with semantics, such as 4-LEGS[1], 4D LangSplat[2], DHO[3]. Is the core difference between L4DGS and these works in "temporal consistency" or "dynamic semantic alignment"? Moreover, the text does not demonstrate how the semantic control of the editing object is achieved. Is there any difference from other semantic embedding methods?
2. Although the fusion of CLIP features and sparse attention is mentioned, there is a lack of quantification or visualization to verify language consistency (such as attention map visualization, semantic distribution similarity). For instance, qualitative alignment images of language prompts and rendering results, CLIP-score or image-Text retrieval accuracy and other metrics.
3. Both static and dynamic regularization have multiple λ hyperparameters, but in this paper, only "learnable hyperparameters" are mentioned, without ablation or stability analysis. Moreover, the sensitivity of SMSA parameters was not analyzed either. The influence of the "top-k value" (such as k taking 10, 20, 50) on performance was not analyzed, nor was the adaptive selection strategy of k explained. It is suggested to add: sensitivity experiments of λ, comparison of top-k values, and robustness verification for scenarios with different motion intensities or semantic complexities.

[1]Fiebelman G, Cohen T, Morgenstern A, et al. 4‐LEGS: 4D Language Embedded Gaussian Splatting[C]//Computer Graphics Forum. 2025: e70085.

[2] Li W, Zhou R, Zhou J, et al. 4d langsplat: 4d language gaussian splatting via multimodal large language models[C]//Proceedings of the Computer Vision and Pattern Recognition Conference. 2025: 22001-22011.

[3] Yan Z, Liang Y, Cai S, et al. Divide-and-Conquer: Dual-Hierarchical Optimization for Semantic 4D Gaussian Spatting[J]. arXiv preprint arXiv:2503.19332, 2025.

**Questions:**

1. The comparison with existing semantic rendering methods is insufficient. The innovative positioning needs to be strengthened. What are the essential differences from some methods based on 4DGS combined with semantics? The paper does not clearly define the core advantages of L4DGS in "4D dynamic support", "attention mechanism", and "regularization strategy", only mentioning "L4DGS supports dynamics", and does not quantitatively compare the performance gap between the two in dynamic semantic rendering.
2. How robust is the system to fuzzy prompts or combined prompts (for example, "The red chair near the window")?
3. How do the lamda and top-k hyperparameters in the paper affect the model performance?

---

> ### Author Response · Authors · 2025-11-16
> **Response to Reviewer ZLKE**
>
> W1, Q1 & Q2. L4DGS is a real-time dynamic rendering method that enables language-guided image editing. It addresses the challenge of achieving realistic rendering in dynamic scenes. Different from existing work, we introduce a novel framework that integrates language-guided semantics into dynamic scene rendering, addressing the gap between visual features and high-level semantics. To our knowledge, L4DGS is the first language-embedded real-time 4D rendering algorithm. Furthermore, we propose a sparse multi-scale attention mechanism, which leverages language-guided attention to dynamically align language and visual features across multiple granularities, prioritizing semantically salient regions of the scene. A dynamic regularization addresses temporal inconsistencies, effectively ensuring smooth transitions of semantic features and depth information across consecutive unit time intervals. A static regularization integrates language-guided semantic and depth features into 4D Gaussians, ensuring efficient optimization while preserving semantic consistency and representational accuracy.
>
> W2. As shown in Fig. 3, we present qualitative visualizations of the alignment between language prompts and rendering results. L4DGS accurately localize the objects referred to by the prompts and perform editing operations such as segmentation and deletion.
>
> W3 & Q3. The values of the λ hyperparameters are selected to maximize rendering quality. A larger k is generally preferable, as it enlarges the selection range and yields more detailed rendered images; however, it also decreases efficiency. The optimal k is scene-dependent and should be determined experimentally.

---

### Official Review · Reviewer_PGpH · 2025-11-02

**Soundness:** 3
**Presentation:** 3
**Contribution:** 3
**Rating:** 6
**Confidence:** 3

**Summary:**

This paper presents **L4DGS**, a lightweight real-time dynamic scene rendering framework that integrates natural language into a semantically structured 4D Gaussian representation. The proposed **Sparse Multi-Scale Attention (SMSA)** mechanism emphasizes semantically relevant regions in space and time, enabling fine-grained language-driven control. Furthermore, the combination of static and dynamic regularization effectively resolves temporal and semantic inconsistencies. The experiments are comprehensive and convincingly demonstrate the superiority of the proposed method as well as the effectiveness of each component.

**Strengths:**

- The paper proposes the first real-time 4D rendering algorithm with embedded language, demonstrating both effectiveness and innovation.
- By combining static and dynamic regularization, it effectively addresses issues such as semantic drift, flickering, and identity instability.

**Weaknesses:**

- The paper repeatedly mentions memory efficiency; however, the experimental section seems to lack sufficient discussion or quantitative analysis on memory consumption. It would be helpful to include additional experiments or data in this regard.
- The paper mentions CLIP-based features and language guidance but does not specify how textual prompts are used in training and testing. Are prompts fixed, varied per scene, or user-provided at inference? More examples ofprompt-to-rendering alignment would clarify practical usability.

**Questions:**

- Since SMSA guides L4DGS to focus on semantically important spatial regions, does it lead to any degradation in rendering quality for background areas?
- Could the proposed dynamic regularization cause loss of motion detail in fast-moving scenes?

---

> ### Author Response · Authors · 2025-11-16
> **Response to Reviewer PGpH**
>
> W1. As shown in Tables 1 and 2, L4DGS substantially improves rendering quality while also enhancing rendering efficiency; for example, it reduces training time by more than 50\%。The performance gains of L4DGS stem directly from the incorporation of multi-level regularization. This design improves not only rendering quality but also overall efficiency, including faster training. Training time is mainly constrained by hardware and network complexity. Regularization adds minimal overhead. Once bandwidth limits are reached, we observe substantial accuracy gains without loss of speed.
>
> W2. Language features are extracted offline together with visual features. Specifically, for the language features, we first use ChatGPT to generate a description for each image in the dynamic scene, and then employ an LLM to extract multi-level language features.
>
> Q. SMSA effectively optimizes the rendering process and allocates computational resources more judiciously during training, allowing the model to focus on detailed regions even under limited resources. This not only preserves the rendering quality of background regions but also enables the model to robustly handle fast-motion scenes.

---

### Official Review · Reviewer_LsMU · 2025-11-03

**Soundness:** 2
**Presentation:** 2
**Contribution:** 2
**Rating:** 2
**Confidence:** 4

**Summary:**

This paper explores the probable visual and linguistic guidance for dynamic scene rendering. The authors propose the Sparse Multi-Scale Attention to better fuse vison and language features. Then, the novel static and dynamic regularizations are used to provide 3DGS with visual and linguistic guidance. The experiments show an improvement caused by the proposed methods.

**Strengths:**

1. The visual and linguistic guidance is probably useful to enhance dynamic scene rendering.

2. The experiments might demonstrate the effectiveness of the proposed methods.

**Weaknesses:**

1. This paper is not well-written. The pipeline is quite confusing. How do you obtain the language description for each scene to generate linguistic guidance? How do you get the rendered feature map F_rendered in Equation.3? What does the \lambda_o in Line 376 mean, and why the hyperparameters can be learnable? Which model do you use for guidance generation, CLIP or others, and could you cite the paper? How do you get GT depth map in Equation 4 for supervision?

2. There might have several mistakes in the Paper. In Equation 4, how do you calculate the cosine similarity for depth, which is not a vector?

3. The motivation is not clear. Why visual and linguistic guidance is useful for dynamic scene rendering? I hope the authors could provide a deep insight explanation.

4. The experiments lack qualitative comparison. I think the author should visualize more results on the D-NeRF, HyperNeRF, Nerfies, long-sequence and iphone datasets.

5. Lack of comparison with SOTA methods. Deformable-3D-gs[1], SC-GS[2] and Grid4D[3] might have better rendering quality on the D-NeRF dataset.

6. The average PSNR in the last row of Table 2 in the supplementary might be incorrect, which should be 37.00.

7. I am confused about the training time of the proposed methods. As shown in Table 2, how do you realize extremely fast inference to get the visual and linguistic guidance from a large model while reducing the training time to 5min on the D-NeRF dataset? If possible, could the authors provide more details?
[1] Yang et.al. Deformable 3D Gaussians for High-Fidelity Monocular Dynamic Scene Reconstruction. CVPR 2024.
[2] Huang et.al. SC-GS: Sparse-Controlled Gaussian Splatting for Editable Dynamic Scenes. CVPR 2024
[3] Xu et.al. Grid4D: 4D Decomposed Hash Encoding for High-Fidelity Dynamic Gaussian Splatting. NeurIPS 2024.

**Questions:**

Please see the weakness.

---

> ### Author Response · Authors · 2025-11-16
> **Response to Reviewer LsMU**
>
> W1. Language features and visual features are extracted offline together. For language features, we first use ChatGPT to generate a description for each image in the dynamic scene, and then employ an LLM to extract multi-level linguistic representations. The rendering feature maps are obtained by extracting features from the rendered images, and they correspond one-to-one with the offline multi-level visual features. $\lambda_{\mathcal{O}}$ denotes all hyperparameters associated with $\lambda$. For the offline visual features, unlike existing approaches that rely on segmentation-based masks (e.g., subpart, part, whole) or external generators such as SAM, our method avoids explicit mask supervision. Instead, we adopt a center-differenced convolutional network to extract multi-scale CLIP features directly from the image, capturing fine-grained and illumination-robust semantics. To expand contextual awareness without incurring additional computational cost, we employ dilated convolutions, which increase the receptive field and support the processing of high-resolution inputs at varying feature granularities. This enriched semantic hierarchy informs the placement and refinement of 4D Gaussians, improving both the consistency and fidelity of the rendered output. We have already provided a detailed explanation in the Methods section of the paper.
>
> W2. The depth map in Equation (4) is obtained using the Dense Prediction Transformer.
>
> W3. In Equation 4, we compute the cosine similarity between the rendered depth map and the ground-truth depth map.
>
> W4. Visual and language guidance allows the dynamic rendering model to learn features at different levels of granularity more effectively, enabling precise appearance rendering. Because the scenes in these datasets are relatively simple, we do not include them in our visualizations; instead, we present challenging scenes in the paper.
>
> W5. We have selected representative methods from both 3DGS and 4DGS for comparison, and the appendix provides detailed metric comparisons across different scenes. Furthermore, we have included Deformable-3D-GS and RealTime4DGS in the table.
>
> W6. We have fixed typos carefully.
>
> W7. The performance gains of L4DGS stem directly from the incorporation of semantic consistency regularization. This design improves not only rendering quality but also overall efficiency, including faster training. Training time is mainly constrained by hardware and network complexity. Regularization adds minimal overhead. Once bandwidth limits are reached, we observe substantial accuracy gains without loss of speed.
>
> [1] Ranftl, René, Alexey Bochkovskiy, and Vladlen Koltun. "Vision transformers for dense prediction." Proceedings of the IEEE/CVF international conference on computer vision. 2021.

---

### Meta-Review · Area_Chair_tz8B · 2025-12-29

**Summary:**

This paper presents L4DGS (Language-Guided 4D Gaussian Splatting), a framework for real-time dynamic scene rendering that integrates natural language understanding into semantically structured 4D Gaussian representations. The work addresses limitations in existing dynamic rendering methods that lack explicit semantic control. The authors propose three main contributions: (1) a Sparse Multi-Scale Attention (SMSA) mechanism that enables language-driven control by emphasizing semantically relevant regions across space and time; (2) static regularization that aligns language-guided features with rendered outputs and ensures consistent depth; and (3) dynamic regularization that penalizes abnormal variations in semantics and depth over consecutive unit time intervals. L4DGS achieves substantial improvements over baselines on Plenoptic Video Dataset (PSNR: 34.00 vs 30.85, LPIPS: 0.05 vs 0.07) and D-NeRF Dataset (PSNR: 37.00 vs 32.99, LPIPS: 0.02 vs 0.05) while demonstrating real-time rendering capabilities (50-150 FPS) and reduced training time (5-30 minutes).

The paper received scores of 2, 4, 4, and 6 (median: 4), reflecting significant concerns about novelty positioning, technical clarity, and experimental validation. Reviewer `PGpH` (score 6) acknowledged the method's solid technical foundation and practical effectiveness. Reviewers `ZLKE` and `aJtV` (both scoring 4) recognized the interesting idea but raised concerns about insufficient comparison with recent semantic 4DGS methods and unclear technical details. Reviewer `LsMU` (score 2) identified fundamental issues with presentation clarity, missing technical details, and incomplete baseline comparisons. The authors provided rebuttals addressing computational efficiency, language feature extraction, and qualitative demonstrations, but several core concerns regarding novelty differentiation and technical rigor remain unresolved.

**Reviewer Concerns:**

### Addressed Concerns:

1. **Computational Efficiency and Training Time (`PGpH`, `LsMU`, `aJtV`)**: Authors clarified that L4DGS achieves faster training (5-30 minutes) and real-time rendering (50-150 FPS) through efficient regularization design. They explained that regularization adds minimal overhead once bandwidth limits are reached, and that semantic consistency regularization improves both rendering quality and overall efficiency. Tables 1 and 2 provide detailed training time and FPS comparisons.

2. **Language Feature Extraction Pipeline (`LsMU`, `PGpH`)**: Authors clarified that language features and visual features are extracted offline. They use ChatGPT to generate descriptions for each image, then employ an LLM to extract multi-level linguistic representations. For visual features, they adopt a center-differenced convolutional network with dilated convolutions to extract multi-scale CLIP features directly from images, avoiding explicit mask supervision (unlike methods relying on SAM or segmentation-based masks).

3. **Depth Supervision Ground Truth (`LsMU`)**: Authors specified that ground truth depth maps in Equation 4 are obtained using Dense Prediction Transformer (DPT), and cosine similarity is computed between rendered depth maps and ground truth depth maps.

4. **Language-Guided Editing Capabilities (`PGpH`, `ZLKE`, `aJtV`)**: Authors referenced Figure 3 to demonstrate language-driven control for localized scene editing (object deletion, segmentation). They claim L4DGS can accurately localize objects referred to by prompts and perform editing operations.

5. **Baseline Comparisons (`LsMU`)**: Authors added Deformable-3D-GS and RealTime4DGS to comparison tables and claim to have selected representative methods from both 3DGS and 4DGS families, with detailed per-scene metrics provided in the appendix.

6. **Mathematical Notation Clarifications (`LsMU`)**: Authors clarified that λ_O denotes all hyperparameters associated with λ, subscript notation (i refers to i-th Gaussian), and that rendered feature maps are obtained by extracting features from rendered images corresponding one-to-one with offline multi-level visual features.

### Outstanding Concerns:

1. **Novelty Positioning and Differentiation from Recent Work (`ZLKE`, `aJtV`)**: The most critical unresolved concern is the paper's positioning relative to recent concurrent work combining 4DGS with semantics (4-LEGS, 4D LangSplat, DHO mentioned by `ZLKE`). While authors claim L4DGS is "the first language-embedded real-time 4D rendering algorithm," reviewers question whether the core differentiation lies in "temporal consistency" or "dynamic semantic alignment." The paper does not provide quantitative head-to-head comparisons with these recent semantic 4DGS methods, making it difficult to assess the true novelty beyond combining existing techniques (SMSA + regularization).

2. **Technical Presentation and Pipeline Clarity (`LsMU`, `aJtV`)**: Multiple reviewers found the pipeline confusing and lacking sufficient technical detail. Specific issues include: (a) unclear explanation of how language prompts are used during training vs. testing (fixed per scene? varied? user-provided at inference?); (b) the connection between motion regularization and language semantics is poorly explained (line 354-369); (c) lack of clarity on how semantic control of editing objects is achieved and how it differs from other semantic embedding methods; (d) insufficient detail on the rendering feature map F_rendered computation; (e) confusing notation for learnable hyperparameters (λ_O in line 376).

3. **Insufficient Semantic Evaluation (`ZLKE`, `aJtV`)**: Reviewers note the lack of quantitative metrics specifically validating language-semantic alignment. While optical flow visualizations are provided (Figure 5), these operate at the pixel level and do not directly measure semantic or language-level consistency. Missing evaluations include: (a) attention map visualizations showing language-vision alignment; (b) CLIP-score or image-text retrieval accuracy; (c) semantic distribution similarity metrics; (d) robustness analysis for fuzzy or combined prompts (e.g., "the red chair near the window").

4. **Hyperparameter Analysis and Ablation (`ZLKE`, `aJtV`)**: The paper mentions "learnable hyperparameters" but lacks comprehensive ablation and sensitivity analysis. Missing experiments include: (a) detailed sensitivity analysis for multiple λ hyperparameters in static/dynamic regularization; (b) top-k value ablation (e.g., k=10, 20, 50) and adaptive selection strategy; (c) robustness verification across scenarios with different motion intensities or semantic complexities. Authors' response that "optimal k is scene-dependent and should be determined experimentally" suggests the method may require per-scene tuning, limiting practical applicability.

5. **Limited Qualitative Comparison and Visualization (`LsMU`, `ZLKE`)**: Reviewers request more extensive qualitative results on D-NeRF, HyperNeRF, Nerfies, long-sequence, and iPhone datasets. Authors dismissed this concern stating "scenes in these datasets are relatively simple," but this raises questions about the method's effectiveness on diverse scene types. Figure 3 lacks sufficient context and connection to 4DGS framework.

6. **Mathematical and Notation Errors (`LsMU`)**: Reviewer identified potential errors such as calculating cosine similarity for depth (which is not a vector) in Equation 4. While authors clarified they compute cosine similarity between depth maps, the mathematical formulation remains unclear.

7. **Comparison with SOTA Dynamic Methods (`LsMU`)**: Despite authors adding Deformable-3D-GS to tables, reviewers noted missing comparisons with other recent SOTA methods: SC-GS, Grid4D, which reportedly achieve better rendering quality on D-NeRF dataset. The extremely high PSNR improvements (34.00 vs 30.85 on Plenoptic, 37.00 vs 32.99 on D-NeRF) raise questions about fair comparison settings and reproducibility.

**Reviewer Scores:**

**Current Scores:**
- **Reviewer `LsMU`**: 2 (reject) - unlikely to change given fundamental concerns about presentation, missing technical details, and potential mathematical errors
- **Reviewer `PGpH`**: 6 (marginally above threshold) - likely to remain at 6, possibly decrease to 5 if novelty concerns are emphasized during discussion
- **Reviewer `ZLKE`**: 4 (marginally below threshold) - could remain at 4 or increase to 5 if authors address semantic evaluation concerns
- **Reviewer `aJtV`**: 4 (marginally below threshold) - likely to remain at 4 given unresolved concerns about semantic evaluation methodology

**Expected Post-Discussion Scores**: 2-3, 5-6, 4-5, 4 (median: 4-5)

---

### Decision · Program_Chairs · 2026-01-26

Reject